

# Brief communication: Enhanced representation of the power spectra of wind speed in Convection-Permitting Models

Nathalia Correa-Sánchez[1], Xiaoli Guo Larsén[2], Giorgia Fosser[3], Eleonora Dallan[1], Marco Borga[1], and Francesco Marra[4]

[1]Department of Land Environment Agriculture and Forestry, University of Padova, Padova, Italy.
[2]Department of Wind Energy, Technical University of Denmark, Roskilde, Denmark.
[3]University School for Advanced Studies - IUSS Pavia, Pavia, Italy.
[4]Department of Geosciences, University of Padova, Padova, Italy.

**Correspondence:** Nathalia Correa-Sánchez (nathalia.correasanchez@phd.unipd.it)

**Abstract.**

The accurate representation of the power spectra of wind speed is crucial for assessing extreme wind speeds, but numerical models often suffer from premature energy loss at high frequencies. Here, we show that Convection-Permitting Models from the CORDEX-FPS can reproduce the theoretical $-5/3$ slope of the 100 m wind speed power spectra in the high frequency range, contrary to other mesoscale simulations used by the wind community (NEWA and ERA5), which exhibit steepened spectral slopes. This superior energy cascade representation is essential for extreme wind estimation and eliminates the need for spectral corrections, opening opportunities for improved wind farm design and more reliable energy transition planning.

## 1 Introduction

A fundamental parameter for wind turbine design is the 10-minute mean wind speed with a 50-year return period (2% annual exceedance probability) at hub height, $U_{50}$. Calculating $U_{50}$ from a wind time series requires the wind variability to be resolved to a temporal resolution of 10 minutes. The power spectrum of wind speed, $S(f)$, as a function of frequency $f$ (in day$^{-1}$), provides a convenient approach to examine this temporal resolution requirement. Larsén et al. (2012) developed a spectral correction method to estimate 10-minute equivalent extreme winds from coarser resolution mesoscale model output. Mesoscale models typically exhibit spectral energy deficits above a cutoff frequency $f_c \approx 2$ day$^{-1}$ compared to the theoretical $-5/3$ slope expected from observations (e.g., Frehlich and Sharman, 2004; Larsén et al., 2013). These spectral limitations are exacerbated by additional biases in reanalysis datasets, such as ERA5, which consistently underestimates strong wind speeds offshore, implying design risks for turbine engineering applications (Gandoin and Garza, 2024).Since the second-order spectral moment $m_2 = \int f^2 S(f)\,df$ weights high frequencies quadratically, these spectral deficits cause substantial underestimation of extreme winds. Therefore, spectral correction methods are used to addresses this issue by replacing the model's deficient spectral tail with the climatologically average spectral slope of $-5/3$ for $S(f)$ versus $f$ in log-log coordinates, extending to the Nyquist frequency of 10-minute data (72 day$^{-1}$). This correction has been used in wind extremes assessment as a post-processing





technique to restore $m_2$ to physically consistent values and improves extreme wind estimates (Bastine et al., 2018; Larsén and Ott, 2022).

Due to the lack of wind measurements at the required height and future projection needs, numerical models are an attractive
alternative in providing data for the extreme wind calculation, as they usually have a good spatial and temporal coverage (global/regional, tens of years). Typically, model simulations are stored every hour (e.g., the reanalysis products such as ERA5, CFSR, and MERRA) or, for some mesoscale regional models, every half an hour or even 10 min (e.g., the New European Wind Atlas, NEWA, based on the Weather Research and Forecasting Model, WRF; Hahmann et al., 2020; Dörenkämper et al., 2020). However, their representation of high-frequency wind variability, and hence extreme wind speed, is often biased.This issue is
exemplified by global extreme wind assessments based on ERA5, where authors acknowledge that coarse model resolution inherently leads to spectral truncation and variance underestimation, pointing out that such corrections could be applied to overcome those field smoothing effects (Pryor and Barthelmie, 2021). Atmospheric models that produce steeper spectral slope than $-5/3$ in the frequency range about 1–72 day$^{-1}$ systematically underestimate both variability and extreme values in the wind speed distribution, compromising the reliability of wind extremes estimation and structural load estimates. It is therefore
important to assess whether these simulations are able to represent the expected high-frequency wind variability, up to the corresponding temporal resolutions required for calculating $U_{50}$.

Skamarock (2004) has shown that the WRF model can resolve a spatial resolution of 7 times the grid spacing $\delta x$ when evaluated at 22-, 10-, and 4-km configurations. However, as spatial resolution is inherently linked to temporal discretisation in model physics schemes, WRF model wind speed output at hourly resolution cannot be expected to reproduce spectral power
at the corresponding high-frequency scales. This temporal limitation is confirmed by spectral analyses in (Larsén et al., 2012), which showed that two other regional models, HIRHAM and REMO, were unable either to reproduce similar spectral behaviour in the frequency range from approximately 1 day$^{-1}$ to 1 h$^{-1}$. It needs to be pointed out that all models in Larsén et al. (2012) employed relatively coarse spatial resolutions ($\delta x$ = 10-50 km). However, even NEWA data generated using WRF at $\delta x = 3$ km exhibits similar spectral deficiencies (Bastine et al., 2018), suggesting that the problem is not only resolution-dependent.

Convection-permitting models (CPMs), developed within the CORDEX Flagship Pilot Studies (FPS) to explicitly resolve convective processes at horizontal grid spacing finer than 3 km and improve the accuracy in representing precipitation (Coppola et al., 2020; Fosser et al., 2024; Ban et al., 2021), provide an opportunity to investigate whether explicit convection also improves the representation of near-surface wind variability at scales critical for extreme value estimation. Despite the increasing use of CPMs in meteorology and climatology, their ability to reproduce wind spectral properties in the context of wind
energy applications has not yet been explored in detail. While some studies suggest that CPMs could better represent kinetic energy spectra compared to lower resolution models (Bierdel et al., 2021; Ricard et al., 2013), the transfer of these advantages to the wind sector remains unexplored. This knowledge gap is particularly relevant considering that an adequate representation of wind variability on multiple time scales is critical for accurate wind resource estimates, prediction of extreme events, and characterisation of intermittency in power generation, all essential elements to adequately prepare for future scenarios.

Here, we examine the power spectra of the wind speed simulated by three CPMs at 100 m height, the typical height of wind turbines, to provide insights into the simulation of high-frequency wind variability for the wind energy sector. We compare the





spectral behaviour of these models against benchmark in situ observations and model simulations commonly used within the wind community to assess whether the CPMs are able to reproduce the theoretical $-5/3$ spectral slope in the high frequency range. We then discuss the practical implications of these findings for wind energy applications, particularly in resource assessment, extremes characterisation and wind farm design.

## 2 Data

### 2.1 Convection Permitting Model simulations

We assess here the three CPMs from the CORDEX Flagship Pilot Project (Coppola et al., 2020) that were publicly available at the time of this research: COSMO-CLM, CNRM-AROME, and COSMO-ETH (Table 1). COSMO-CLM is a regional climate model commonly used also for high-resolution simulations (Adinolfi et al., 2020), while CNRM-AROME employs parameterisations optimised for Mediterranean systems (Caillaud et al., 2021). COSMO-ETH, similar to COSMO-CLM, offers a GPU-accelerated implementation for continental-scale simulations (Leutwyler et al., 2017, 2016). Though all operate at similar convection-permitting resolutions ($\leq$3 km) and are non-hydrostatic (allowing explicit vertical accelerations), they differ in numerical discretisations, physical parameterisations and diffusion formulations (Coppola et al., 2020), all of which may influence the spectral energy distribution (Malardel and Wedi, 2016). The 100 m wind speeds were calculated directly from the zonal (U) and meridional (V) wind components simulations available at this level without requiring vertical extrapolation. All CPMs are nested in a European 12 km domain driven by ERA-Interim. Additionally, the simulations are available at hourly frequency and cover the period 2000-2009. The specifications are listed in Table 1 for each CPM member, and the common domain covered by the CPM members is shown in Fig. A1a.

### 2.2 Observational data

The observed wind speed data at 100 m height used in this study were provided by the Institute for Meteorology and Climate Research - Troposphere Research (IMK-TRO) of the Karlsruhe Institute of Technology (KIT). These records come from the 200 m meteorological tower located at the KIT North Campus (49.0925°N, 8.4259°E, 110.4 m above sea level; see Fig. A1a), which has been continuously operational since 1972 and is equipped with high-precision instruments for turbulence measurement. The 100 m wind speed data from the KIT mast cover the period 2000-2009, being the only open-access records available within our study domain, upon specific request to the institute. Originally recorded at 10-minute intervals, these data were subsequently aggregated to hourly values by arithmetic averaging to facilitate direct comparison with the simulations of the CPM models. The quality and completeness of the data were checked prior to sampling for each year before aggregating the time series.





**Table 1.** List of the three CPM members with technical specifications, including coupled RCM information.

| Institute | CPM | Numerical Discretisation | Horizontal Diffusion | RCM |
|---|---|---|---|---|
| **CMCC** | CCLM | Finite differences | 4th-order Smagorinsky hyper-diffusion | CCLM |
| Euro-Mediterranean Center on Climate Change | 3 km (Adinolfi et al., 2020) | 3rd-order Runge-Kutta, 5th-order upwind advection | | 12 km (Adinolfi et al., 2020) |
| **CNRM** | CNRM-AROME41t1 | Bi-spectral ALADIN core | Semi-Lagrangian horizontal diffusion (SLHD) | CNRM-ALADIN63 |
| Centre National de Recherches Météorologiques | 2.5 km (Caillaud et al., 2021) | Semi-implicit discretisation | | 12 km (Nabat et al., 2020) |
| **ETH** | COSMO-crCLIM | Finite differences | No explicit horizontal diffusion | COSMO-crCLIM |
| Institute for Atmospheric and Climate Science | 2.2 km (Leutwyler et al., 2016) | 2-timelevel 3rd-order Runge-Kutta, 5th-order upwind | | 12 km (Leutwyler et al., 2017) |

## 2.3 New European Wind Atlas

Focusing on the location of the KIT mast, for a direct comparison with the available observed data, wind speed time series at 100 m from the New European Wind Atlas (NEWA) were extracted for a time period consistent with that of the CPM models and observational data, i.e. 2000-2009. Hahmann et al. (2020) describe the sensitivity simulations performed to select the optimal mesoscale WRF model configuration used in the generation of NEWA, which is a set of WRF climate simulations using spectral nudging (different from the freely evolving CPMs from CORDEX-FPS). Although the internal domain is also 3 km, the NEWA simulation uses spectral nudging in the outer domain (27 km), which constrains the large-scale flow influencing the 3 km simulations. The specific diffusion-related numerical settings that Hahmann et al. (2020) applied to the WRF model simulations for NEWA are detailed in their annexes and are based on 'best practice' guidelines and modeller's experience, with the aim of maintaining numerical stability and controlling errors associated with discretisation in mesoscale simulations—especially important at the 3 km resolution of the inner domain. The global evaluation of NEWA against 291 meteorological masts supports the performance of the final configuration selected for the atlas in terms of mean wind speed and wind direction statistics at 100 m height (Dörenkämper et al., 2020). However, this evaluation did not specifically address the spectral characteristics or the representation of high-frequency variability that we examine in this study.





## 2.4 ERA5

Wind speed data at 100 m height from the ERA5 reanalysis were calculated directly from the U and V components available at this level from the Climate Data Store (Hersbach et al., 2020) for the location of the KIT mast, and constitutes an hourly time series covering the period 2000-2009 at a spatial resolution of $0.25° \times 0.25°$ (approximately 31 km at this latitude). The relevance of ERA5 in our study is that it combines global observations with atmospheric physical models through the ECMWF data assimilation system, providing consistent estimates of the atmospheric state. ERA5 is the successor of ERA-Interim, used as forcing in the CPMs, among the reanalyses developed by the ECMWF. This data source complements the set of observations and diverse model simulations analysed in this study, allowing for a comprehensive assessment of the spectral characteristics of the wind speeds. The inclusion of ERA5 in our comparative analysis provides insights into the representation of wind variability from different schemes and approaches, since this reanalysis product is widely used in climate research and wind energy applications.

## 3 Methods

Here we use a comparative approach to examine whether the raw spectra from all three CPMs exhibit energy deficits at high frequencies that affect wind variability representation, a limitation previously attributed to artificial spectral damping in numerical models (Wang and Sardeshmukh, 2021; Skamarock, 2004). We compare the CPM spectra with both simulations' spectra corrected with the methodology of Larsén et al. (2012) and observed spectra to determine whether continuous CPMs' free-running simulations reduce, maintain, or exacerbate these high-frequency deficits compared to nudged high-resolution models. For this, the hourly time series of CPMs, ERA5, and NEWA, were first detrended by subtracting their mean value, thus removing the constant component.

Subsequently, we applied Welch's method with Hanning windowing to obtain the power spectral density (PSD), using segment lengths of 1024 data with 50% overlap between segments. This method divides the time series into overlapping segments, applies a windowing function to each segment, then computes the FFT, and averages the resulting periodograms to reduce spectral variance while maintaining frequency resolution. Here, frequencies are expressed in days$^{-1}$ following standard practices in atmospheric spectral analysis (e.g., Larsén et al., 2012; Skamarock, 2004).

The cutoff frequency $f_c$ and its corresponding spectral value $S(f_c)$ were determined by a linear regression fit to the doubly logarithmic transformation of the spectrum in the frequency range 0.6-0.9 days$^{-1}$, as recommended by Larsén et al. (2012) to adequately capture the transition between the spectral regimes. In this sense, we make an explicit comparison between the raw and corrected spectra. Following Larsén et al. (2012), the spectral correction methodology is implemented for the time series in the KIT mast point. The corrected spectra for high frequencies are calculated as:

$$S(f)_{\text{corr}} = S(f_c) \cdot (f/f_c)^{-5/3} \quad \text{for} \quad f \geq f_c \tag{1}$$

where $-5/3 \ (\approx -1.67)$ represents the theoretical slope expected from energy cascade processes in the mesoscale range.





We first focused on the location of the KIT mast as a specific validation reference point, where direct wind speed measurements at 100 m height are available, and we extracted the points geographically closest to this from the CPMs, NEWA and ERA5. This allows us to validate CPMs, NEWA, and ERA5 simulations against the observational data and provide a direct and reliable reference to assess the realism of the spectra generated by the CPMs, particularly at the high frequencies where the theoretical behaviour slope of $-5/3$ is expected. Furthermore, since previous studies have already demonstrated the spectral limitations of NEWA and ERA5 (Bastine et al., 2018; Wang and Sardeshmukh, 2021) and to ensure that our results were not driven by a fortuitous choice of the benchmark location, we selected 10 locations within the study domain characterised by diverse topographic and climatic settings (we sampled a range of elevations including both marine and terrestrial environments, see Fig. A1). This approach allows assessing whether the spectral properties of CPMs show spatial consistency or systematic variations related to terrain features.

## 4 Results and Discussion

Figure 1a-c shows the wind power spectra at the KIT mast from the three CPMs and the observations. Remarkably, all raw CPMs spectra closely follow the theoretical slope of $-5/3$ at the high frequencies of the mesoscale spectral range (pale lines in the background). In order to give a visual comparison of what the corrected spectra would look like, we also applied the spectral correction of Larsén et al. (2012) (solid lines). As can be seen in Fig. 1, the corrected and raw spectra from the CPMs are almost identical, confirming that CPMs already represent the spectral behaviour at high frequencies. The CMCC model, however, shows slightly larger divergence between its raw and corrected spectra, suggesting relatively more artificial spectral damping.

On the other hand, Fig.1d shows the wind power spectra from the NEWA and ERA5 data at the KIT measurement site. In both cases, the difference between the raw and corrected spectra is substantial, as the raw power spectra slope is steeper than the reference $-5/3$ slope and with a marked difference from the corrected spectra, evidencing the divergence in the energy cascade representation. This result reaffirms that this type of data need to undergo spectral correction techniques in order to capture high-frequency variability.

To assess whether this improved spectral performance represents a systematic feature rather than an isolated case, Fig. 2 displays the raw power spectra for the three CPM models at 10 additional locations (Series 1-10). Consistent with Fig.1a-c, the power spectra align with the theoretical reference lines in the high-frequency region ($-5/3$ slope) across all locations. This agreement indicates that the CPM's realistic representation of the spectral characteristics is not an isolated phenomenon at the KIT mast location, but rather represents an inherent property of 100 m wind simulations in CPMs. At the same time, systematic differences in energy levels are observed between the three models, with the ETH model generally showing the highest energy, followed by CMCC and CNRM. This is related to the large-scale offsets between the different models. While these energy level differences affect the absolute magnitude of wind variability estimates, the consistent preservation of the $-5/3$ slope across all models demonstrates that the fundamental spectral features remain intact regardless of specific model implementation. Nevertheless, all models share technical inherent limitations. Figures 1 and 2 exhibit a flattening of the spectral slope at the

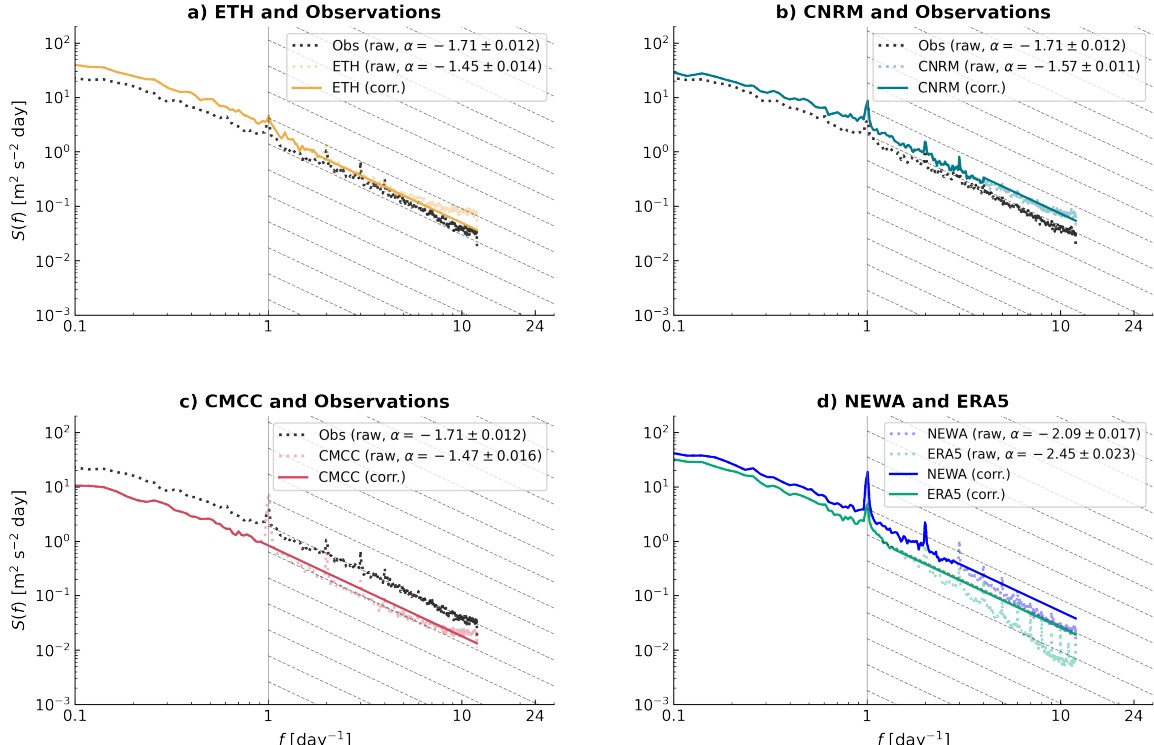

**Figure 1.** Power spectra of the wind speeds at the measurement mast location for the 3 CPMs and the observed data ('Obs'): a) ETH, b) CNRM, c) CMCC. d) The ERA5 and the NEWA power spectra. Pale coloured lines show the smoothed raw spectra for each simulation dataset. In contrast, the solid lines of corresponding colours represent the spectra after applying the spectral correction method proposed by Larsén et al. (2012). The grey dashed lines in the high-frequency region indicate the theoretical -5/3 slope. $\alpha$ denotes the slope of the linear fit with the associated uncertainty expressed as a $\pm$ value.

highest frequencies ($f > 8$ day$^{-1}$) across all simulation datasets, which likely indicates that the models are approaching the temporal resolution limit, where numerical predictions cease to represent physical processes and likely begin to show numerical
artifacts.

The spatial consistency of spectral properties suggests that CPMs can serve as reliable tools for evaluating potential wind farm sites, which is evident even in mountainous terrain, without needing additional spectral corrections. This may drastically simplify the process of obtaining the peak/extreme values. In addition, the enhanced representation of the energy spectra, particularly at high frequencies, suggests that these CPMs can provide more accurate estimates of extreme wind speeds. Since
extremes depend primarily on the tail of the probability distribution, realistic spectral behaviours help to reduce uncertainty in the estimation of extreme events relevant to turbine structural design. These results pave the way for the use of CPMs for





**Figure 2.** Power spectra of wind speed time series from three CPMs at 10 randomly selected locations (a-j). Pale coloured lines show the smoothed raw spectra for each model. The grey dashed lines in the high-frequency region indicate the theoretical -5/3 slope. $\alpha$ denotes the slope of the linear fit with the associated uncertainty expressed as a $\pm$ value.

future projections of extreme wind speed, facilitating the integration of wind energy into electricity grids through more accurate prediction models.

The differences in spectral behaviour between NEWA, ERA5, and direct measurements illustrate how the enhanced spectral characteristics in the CORDEX-FPS CPMs can contribute to reducing uncertainties in energy planning. On the other hand, the improved spectral performance of these CPMs compared to other high-resolution datasets such as NEWA (Fig. 1d) reveals that convection-permitting resolution alone is insufficient to guarantee realistic spectral characteristics. While both NEWA and CORDEX-FPS operate at similar spatial resolutions ($\leq 3$ km), and both resolve convection explicitly, their different numerical configurations produce markedly different spectral behaviours. NEWA employs spectral nudging to maintain consistency with





reanalysis forcing (Hahmann et al., 2020), which constrains large-scale patterns but may interfere with the natural energy cascade processes that generate mesoscale variability. In contrast, the CORDEX-FPS models examined here use freely evolving simulations that allow convective processes to develop their intrinsic spectral characteristics without large-scale constraints (Coppola et al., 2020). This fundamental difference in the simulation approach may explain why these CPMs naturally preserve the energy sources and cascades that produce realistic high-frequency wind variability, whereas constrained high-resolution

models require spectral correction methods to compensate for artificially steepened spectra.

These findings contrast with well-documented limitations in traditional mesoscale approaches. Larsén et al. (2012) identified the numerical smoothing effect resulting in low spectral energy at high frequencies, which systematically affects extreme wind estimation. Olsen et al. (2017) confirmed these limitations using mainly the WRF model in an intercomparison of 25 configurations on simple terrain, while Vincent and Hahmann (2015) demonstrated that nudging techniques significantly reduce

wind variance on mesoscale scales.

Recent studies of current reanalysis and NWP datasets reveal persistent spectral deficiencies. Wang and Sardeshmukh (2021) found "highly inconsistent" mesoscale kinetic energy spectra across global reanalysis products (ERA-Interim, JRA-55, ERA5, NOAA GFS), with energy differences reaching factors of 47 at spatial scales smaller than 400 km. All exhibited steeper slopes than the theoretical $-5/3$ in mesoscale ranges, attributed to inadequate data assimilation at small scales and scale-dependent

numerical damping.

The CPMs evaluated in this study seem to have overcome these fundamental limitations. While convection-permitting resolution provides the necessary spatial scales, simulation designs prioritising physical process fidelity over climatological constraints, seem to be crucial for preserving realistic wind variability characteristics despite the computational costs involved. The implications extend beyond spectral accuracy to practical applications, as these enhanced characteristics in CPMs can provide

more reliable estimates of extreme winds and turbine fatigue loads without requiring post-processing corrections, facilitating more accurate wind energy assessments and grid integration planning since CPM simulations also include projections for the near and far future.

## 5   Conclusions

We examined the power spectra of wind speed of three Convection-Permitting Models from the CORDEX-FPS initiative. The

examined CPMs can reproduce the high-frequency behaviour in the 1–12 day$^{-1}$ range with slopes approaching the theoretical $-5/3$ expectation. This contrasts with previously documented energy deficits in this frequency range observed in other datasets, including simulations with similar resolutions, which were subject to significant energy loss in this frequency range. Indeed, simulations from a global reanalysis product (ERA5) and from a convection-permitting model specifically tailored to wind applications (NEWA) exhibit significantly steeper spectral slopes (-2.45 and -2.10, respectively), requiring post-processing

corrections to achieve realistic wind variability characteristics.

These fundamental differences in spectral performance could be related to the contrasting simulation philosophies rather than spatial resolution alone. While the three CPMs and NEWA datasets operate at convection-permitting scales ($\leq$3 km),





NEWA employs spectral nudging and frequent restarts (36-hour runs). On the other hand, ERA5 uses data assimilation with systematic temporal discontinuities, both of which interfere with natural energy cascade processes. In contrast, CORDEX-
FPS CPMs operate as continuous and freely evolving climate simulations that preserve the downscale energy transfer from large-scale motions to mesoscale variability. Furthermore, CPMs simulations are specifically optimised to accurately represent convective processes at intraday timescales ($<$ 24 hours), which directly correspond to the high-frequency domain ($1\text{--}12\,\mathrm{day}^{-1}$) where spectral improvements are observed.

The superior spectral representation achieved by CPM simulations eliminates the need for post-processing techniques such
as the spectral correction method, simplifying the analysis methodology for estimating the turbine design parameter $U_{50}$. This improvement is expected to translate into more accurate estimates of extreme wind speeds and turbine fatigue loads, as realistic high-frequency variability directly affects the calculation of spectral moments that govern extreme value statistics. Moreover, since these CPMs are specifically developed for future climate projections, our results open the way to a direct use of simulations for wind resource assessment and extreme wind speed quantification under future climate scenarios, supporting
more reliable energy transition planning.

*Code availability.*  Scripts are available by contacting NCS.

*Data availability.*  The CPM data used in this study can be retrieved from the ESGF data node server. The observational data cannot be shared by the authors, but can be obtained from the data owner (*https://www.imk-tro.kit.edu/english/7779.php*) upon request.

**Appendix A:  Location of the randomly selected points**

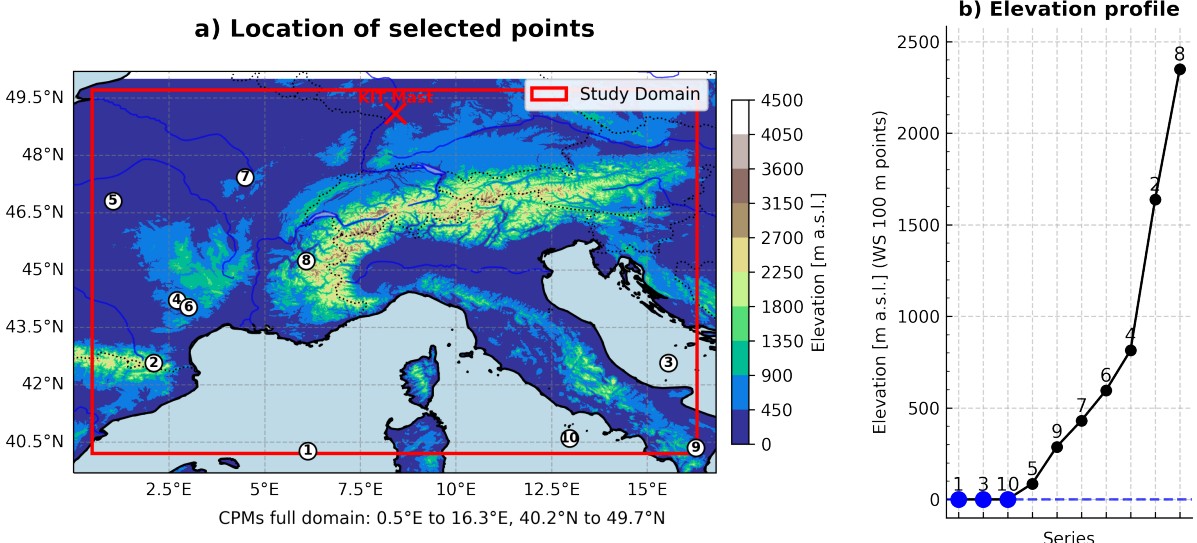

**Figure A1.** a) Location of selected random points in white background circles with the indicated number of the series at each point. Elevations (m a.s.l.) are from ETOPO 2022 (MacFerrin et al., 2025) remapped to 3 km. The red cross indicates the location of the mast of the KIT observations. Marine areas are in light blue. b) Elevation profile above sea level of the series of all random points, the maritime points are marked in blue.



*Author contributions.* NCS, Data curation, Formal analysis, Software, Visualisation, Writing (original draft preparation); NCS, XGL, FM Conceptualisation, Methodology; NCS, XGL, GF, FM Investigation; XGL, FM Supervision; MB Funding acquisition; NCS, XGL, GF, ED, MB, FM Writing (review and editing).

*Competing interests.* The authors declare that they have no conflict of interest.

*Acknowledgements.* This study was partially supported by the CARIPARO Foundation through the Excellence Grant 2021 to the "Re-
silience" Project. FM was supported by the "The Geosciences for Sustainable Development" project (Budget Ministero dell'Università e della Ricerca–Dipartimenti di Eccellenza 2023–2027 C93C23002690001). ED was supported by the RETURN Extended Partnership and received funding from the European Union Next-Generation EU (National Recovery and Resilience Plan – NRRP, Mission 4, Component 2, Investment 1.3 – D.D. 1243 2/8/2022, PE0000005). XL acknowledges support from Horizon Europe DTWO project (101146689).



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
