# Peer review of "Brief communication: Enhanced representation of the power spectra of wind speed in Convection-Permitting Models"

_Wind Energy Science, 2025_

## Referee Comment (RC1)

**Review of "Enhanced representation of the power spectra of wind speed in Convection-Permitting Models"**

**General comments**

The authors present an analysis of the wind speed power spectra from four regional convection-permitting models (three from the CORDEX Flagship Pilot Project and the New European Wind Atlas, NEWA). The wind speed power spectra are compared with theoretical expectations, and an observed spectrum from mast data, and contrasted with a reanalysis dataset that parameterizes convection (ERA5). The key finding of the work is that the convection-permitting models from the CORDEX project produce a wind speed power spectrum for high frequency variations (sub-daily) that matches with observations, offering an improved representation compared with the NEWA and ERA5. This has implications for wind resource and extreme wind assessments using model datasets. The improvement in representation over the NEWA is suggested to be a result of not using large-scale nudging in the model configuration. Although, this is not shown in the paper, and I have a comment related to this theory, below.

Overall, this a nice analysis that is well-written, and I think it is suitable for publication in Wind Energy Science after minor revisions (see comments below).

**Specific comments**

1. The authors have demonstrated the improved power spectrum of wind in the CORDEX CPMs. It is suggested that this is a result of the models being free-running and not nudged towards the large-scale. But I wonder if this improvement is balanced by other aspects becoming less accurate than the nudged simulations. What about the wind speed distribution, for example? This will also affect estimations of $U_{50}$. There are some biases in the CORDEX model wind speed suggested in Figure 1, as acknowledged by the authors. I know the authors may not be able to provide any further analysis of this in this brief communication, but maybe this could be mentioned or relevant references cited?

2. Line 5: I don't think ERA5 is a "mesoscale" model

3. Some more background could be useful in the Introduction. For example, I am not sure what the second-order spectral moment is, and how it relates to estimating extreme winds. Same for Nyquist frequency on line 20.

4. The authors say that hourly wind data is used, but do not mention how the temporal window of these data are defined. For example, do the wind speeds represent 10-minute averages, hourly averages, or instantaneous wind speeds at the hourly model time step? I imagine that this could impact the high frequency variations assessed in this study, especially if they differ between datasets.

5. Figure A1: I assume the CPMs cover a much larger area than this? It would be useful to see the points in relation to the CPM domain (in case any points are close to the boundary, for example).

6. Figure 1: Could $f_c$ be indicated to show where the slopes have been corrected? And why aren't the observations shown on panel d?

7. Line 116: I was slightly confused by "de-trended" as I usually would interpret this to mean that a long-term trend was subtracted from the time series. But it sounds like the wind speed anomalies from the mean were calculated?

8. Line 145: Is "almost identical" a correct assessment? The two lines appear to diverge significantly at the upper tail.

9. Do the spectrum correction methods preserve the diurnal and semi-diurnal wind cycle? Some of these peaks appear to disappear in Figure 1 (but not all). This is not crucial to the paper, but I am curious.

10. Line 159: What is meant by large-scale offsets? I assume this relates to bias in the wind speed distribution in the models, including the mean wind speed?

**Technical corrections**

1. Acronyms should be defined throughout where they first appear, such as on lines 4 and 5. There are some acronyms that are not defined (HIRHAM, REMO, CFSR, MERRA)

2. Line 20: Is 'climatological average' the correct term here?

3. Line 91: Should be "3 km grid spacing"

---

## Author Comment (AC1)

**RESPONSES POINT BY POINT TO REFEREE N°1 OF WIND ENERGY SCIENCE DISCUSSIONS**
*Brief communication: Enhanced representation of the power spectra of wind speed in Convection-Permitting Models*
Nathalia Correa-Sánchez, Xiaoli Guo Larsén, Giorgia Fosser, Eleonora Dallan, Marco Borga,
and
Francesco Marra

Dear Referee No.1,

We thank you for your review work and the valuable comments, which helped to improve our paper. Our responses are reported in blue, and all the modified or new text is reported in *italics and red*. Line numbering refers to the original version of the paper that was available for the open discussion.

**General comments**

The authors present an analysis of the wind speed power spectra from four regional convection-permitting models (three from the CORDEX Flagship Pilot Project and the New European Wind Atlas, NEWA). The wind speed power spectra are compared with theoretical expectations, and an observed spectrum from mast data, and contrasted with a reanalysis dataset that parameterizes convection (ERA5). The key finding of the work is that the convection-permitting models from the CORDEX project produce a wind speed power spectrum for high frequency variations (sub-daily) that matches with observations, offering an improved representation compared with the NEWA and ERA5. This has implications for wind resource and extreme wind assessments using model datasets. The improvement in representation over the NEWA is suggested to be a result of not using large-scale nudging in the model configuration. Although this is not shown in the paper, and I have a comment related to this theory, below. Overall, this a nice analysis that is well-written, and I think it is suitable for publication in Wind Energy Science after minor revisions (see comments below).

Thank you very much for your positive comments about our work. We will respond to each of your specific comments in the following section.

**Specific comments**

**Comment #1.** The authors have demonstrated the improved power spectrum of wind in the CORDEX CPMs. It is suggested that this is a result of the models being free-running and not nudged towards the large-scale. But I wonder if this improvement is balanced by other aspects becoming less accurate than the nudged simulations. What about the wind speed distribution, for example? This will also affect estimations of U50. There are some biases in the CORDEX model wind speed suggested in Figure 1, as acknowledged by the authors. I know the authors may not be able to provide any further analysis of this in this brief communication, but maybe this could be mentioned or relevant references cited?

We thank the reviewer for this clarification. We agree that our study focuses specifically on wind speed variability over short time frames, from a few hours to about one hour. This is one of the

crucial factors for correctly calculating U50. However, we acknowledge that we are not examining all the factors that can influence the accuracy of U50 estimation.

Relevant literature indicates that improved wind speed data at high frequencies directly enhances the estimation of extreme wind speed (Larsén et al., 2012; Bastine et al., 2018), since realistic high-frequency variability is essential for calculating the spectral moments that are key to extreme value statistics. Yet, U50 accuracy relies on multiple factors beyond spectral features, including the accuracy of wind speed distribution, biases in mean wind speed, and the methods used for extreme values.

While the CPMs display biases in Figure 1, our analysis shows that their improved spectral quality eliminates the need for applying additional post-processing corrections typically required for nudged simulations, such as NEWA, and reanalysis products, like ERA5. This provides a significant advantage for wind energy applications.

We will clarify in the manuscript that our study addresses one critical aspect of U50 estimation: the representation of high-frequency wind variability, while acknowledging that a further validation of all wind energy factors is important for future research.

We will add this clarification in the introduction Line 54: *"Here, we specifically study wind speed variability at temporal scales from a few hours to ~1 hour, which is crucial for accurate U50 estimation as high-frequency variability directly affects extreme value statistics (Larsén et al., 2012), but this brief communication does not address all factors that influence extreme wind speed estimation accuracy in different modelling approaches.*

*We examine the power spectra of the wind speed simulated by three CPMs at 100 …".*

And we will also highlighted this issue at the end of the Discussion section in Line 203:

*"While literature indicates that enhanced spectral characteristics directly improve extreme wind estimates through spectral correction methods (Bastine et al., 2018; Larsén and Ott, 2022), further validation across wind speed distributions, mean wind biases, and extreme value methodologies represents an important research priority for wind energy applications."*

Moreover, thanks to the review made for this comment, we realised that Line 213 of the current text should be corrected to state that *" …NEWA employs multi-day simulations with spectral nudging (8-day runs). On the other hand… "* rather than *frequent restarts every 36 hours*, to clearly show the methodological differences between the above sets which, according to previous literature, influence the spectral characteristics of winds.

**Comment #2.** Line 5: I don't think ERA5 is a "mesoscale" model.

We thank the referee for pointing out that we need to make this clearer in the abstract. We recognise that ERA5 is a global reanalysis product, not a mesoscale model. Line 5 should be changed to reflect this difference. We are going to correct the text in the abstract (lines 5 and 6) with: *"…contrary to other mesoscale simulations and global reanalysis used by the wind community (NEWA and ERA5, respectively), which exhibit steepened spectral slopes…"*

We appreciate the reviewer for this clarification about the correct classification of atmospheric products.

**Comment #3**. Some more background could be useful in the Introduction. For example, I am not sure what the second-order spectral moment is, and how it relates to estimating extreme winds. Same for Nyquist frequency on line 20.

We appreciate the referee's suggestion. Although the manuscript already defines the second-order spectral moment $m2$ and mentions the Nyquist frequency (lines 17-21). We have now added some explanation for explicit clarity. First, we will add, in the manuscript version, line 19: "*…cause substantial underestimation of extreme winds. This moment quantifies the contribution of short-timescale fluctuations to total wind variance, and is particularly relevant for estimating wind extremes (Frehlich & Sharman, 2004; Larsén et al., 2012). Therefore, spectral correction methods are…*" to improve the conceptualisation of the second-order moment. Then, we will add the following to line 21 of the original manuscript: "*...extending to the Nyquist frequency of 10-minute data (72 day^-1). The Nyquist frequency represents the maximum resolvable frequency given the data's sampling interval; for hourly and 10-minute time series, this corresponds to 12 and 72 day^-1, respectively (Skamarock, 2004; Larsén et al., 2012).*"

**Comment #4.** The authors say that hourly wind data is used, but do not mention how the temporal window of these data are defined. For example, do the wind speeds represent 10-minute averages, hourly averages, or instantaneous wind speeds at the hourly model time step? I imagine that this could impact the high-frequency variations assessed in this study, especially if they differ between datasets.

We thank the referee for pointing this relevant detail about the time window definitions of our datasets. We provide the following specifications for each dataset:

• CORDEX-FPS CPMs: Instantaneous hourly values showing the model state at each hourly timestep.

• ERA5: Instantaneous values (with an implicit 30-minute average of elapsed time).

• NEWA: Output at 30-minute temporal resolution (Hahmann et al., 2020), aggregated to hourly values for this analysis.

• Observations: Initially recorded every 4 seconds and aggregated to 10-minute means (Kohler et al., 2018), later combined into hourly mean values for comparison.

However, it is essential to note that our study examines how each dataset performs in the relevant frequency range, spanning from a few hours to ~ 1 hour, for spectral analysis. Here, we do not use this data for direct extreme wind calculations, as we stated in the introduction. At these temporal scales, different definitions of temporal windows do not greatly change our main conclusions about spectral slope behaviour.

This is because the processes that create the theoretical -5/3 slope in the mesoscale range (1-6 day^-1) work at characteristic timescales of about 4-24 hours that are much longer than the differences between instantaneous hourly values and 10-minute averages. Additionally, we

confirmed this by comparing the spectral slopes between the closer 10-minute observational data to the hour and the hourly-averaged values, and found no significant differences in spectral behaviour at frequencies of ≥1 hour^-1, confirming that our comparative spectral analysis remains valid for the frequency range of interest in our study.

Therefore, while we acknowledge these methodological differences, they do not impact our primary findings regarding the superior spectral representation in CPMs compared to nudged simulations, such as NEWA, and reanalysis products, like ERA5. In this sense, we will aggregate the following paragraph in line 110:

*"All datasets are analysed at an hourly frequency for consistency and comparison, but they have different time window definitions. CPMs represent instantaneous model states. ERA5 offers instantaneous values, averaging them over 30 minutes. NEWA provides 30-minute resolution data, grouped into hourly values. Observations are collected from 10-minute averages and converted to hourly values. These differences in time windows do not influence spectral analysis in our frequency range of interest, which is from 1 to 6 $day^{-1}$, because the atmospheric processes that generate the -5/3 spectral slopes operate at much longer timescales (4-24 hours) than these methodological differences."*

**REFERENCE:** Kohler, M., Metzger, J., & Kalthoff, N. (2018). Trends in temperature and wind speed from 40 years of observations at a 200-m high meteorological tower in Southwest Germany. *International Journal of Climatology*, *38*(1), 23-34.

**Comment #5**. Figure A1: I assume the CPMs cover a much larger area than this? It would be useful to see the points in relation to the CPM domain (in case any points are close to the boundary, for example).

We appreciate this comment. The CPM simulations do indeed cover a wider domain (the total extension of the elevation model shown in Figure A1a) than the one outlined in red in Figure A1a. This red box was conservatively defined within the full CPM domain to avoid boundary effects in the selection of evaluation points. The points themselves were generated through a spatially uniform random sampling within this reduced area to minimise spatial bias. As a result, some points appear close to the red box edges, though still well within the CPM domain and within the margin to avoid edge effects.

We also note that the plotted markers/tags are larger than the actual CPM grid cells (3 km × 3 km), which can visually exaggerate proximity to boundaries. We will improve the description of what we define as 'Study Domain' in the caption of Figure A1a: *"Study Domain: is an internal domain within the total extent of the CPMs, which has been established to avoid edge effects in the random selection of points. Note that the label of the selected locations has a visible size but exceeds the 3 km x 3 km spatial resolution."*

**Comment #6.** Figure 1: Could fc be indicated to show where the slopes have been corrected? And why aren't the observations shown on panel d?

Thank you for your comment. However, during the development of the research, although the *fc* is calculated, it was decided to remove it from the graphs because the intention is to preliminarily illustrate spectral correction, as a general context in model data, but not to open a debate on the cut-off frequency (*fc*),  since in this discussion it is not relevant and the focus of attention on the subject matter could be lost, in addition to exceeding the scope and extent of the work.

On the other hand, the observations in panel d were not included because they were intended to be used as a reference for the three CPMs (a-c), since previous studies had already shown that ERA 5 and NEWA have an energy deficit at high frequencies, so this is not a new finding.

For clarity, we will add to line 129: *"...and fc is the frequency where the slope deviates from the theoretical one."*

**Comment #7**. Line 116: I was slightly confused by "de-trended" as I usually would interpret this to mean that a long-term trend was subtracted from the time series. But it sounds like the wind speed anomalies from the mean were calculated?

We thank and agree with the referee's comment about the terminology. The referee is correct that 'detrended' is usually used to remove low-frequency oscillations and long-term trends over time. In our analysis, we calculated wind speed anomalies by subtracting the average of the time series. We do this to centre the data around zero and remove the constant component needed for spectral analysis while preserving the temporal trend. We will change the term in line 116 to *'mean-centred'* to better describe the preprocessing done before spectral analysis.

**Comment #8**. Line 145: Is "almost identical" a correct assessment? The two lines appear to diverge significantly at the upper tail.

We appreciate the comment and have corrected the text from the current lines 144 and 146 to adequately describe what is presented in the results of Figure 1, explaining concisely but rigorously the causes of the divergences in the very high frequency limits (frequencies > 10 day^-1). The text will be modified as follows:

*"…As can be seen in Fig. 1, the corrected and raw spectra from the CPMs show good agreement in the mesoscale frequency range (1-10 day^-1), confirming that CPMs adequately represent spectral behaviour at these frequencies. However, divergence occurs at frequencies > 10 day^-1 (periods < 2.4 hours), approaching the effective temporal resolution limit of hourly model output, where the representation of sub-daily atmospheric variability becomes increasingly uncertain.  The CMCC model, however, shows slightly…"*

**Comment #9**. Do the spectrum correction methods preserve the diurnal and semi-diurnal wind cycle? Some of these peaks appear to disappear in Figure 1 (but not all). This is not crucial to the paper, but I am curious.

We thank the reviewer for their interest in the diurnal and semi-diurnal cycles. The cycles are visible in the raw spectra, which is consistent with the expected atmospheric behaviour.  On the other hand, the preservation of diurnal (f = 1 day^-1) and semi-diurnal (f = 2 day^-1) cycles in spectral correction relies, indeed,  on the choice of cutoff frequency (*fc*) relative to these characteristic frequencies. When *fc* is set below these frequencies, the cycles are preserved.

When *fc* is above them, the correction may smooth these peaks, as noted by Larsén et al. (2022) (https://doi.org/10.1002/we.2771).

**Comment #10**. Line 159: What is meant by large-scale offsets? I assume this relates to bias in the wind speed distribution in the models, including the mean wind speed?

Thank you for asking us to clarify this term. By "large-scale offsets," we mean the clear differences in absolute spectral energy levels among the three CPM models. These differences appear across all frequencies and locations (Figures 1 and 2). As you correctly assume, they show how the models represent wind speed distributions, including mean wind speeds and overall variability. The ETH model consistently has the highest spectral energy levels, followed by CMCC and CNRM.

These differences come from unique model setups, such as different parameterisation schemes, boundary layer representations, and numerical discretisations used inside each CPM. While these energy level differences impact the absolute size of wind variability estimates, the main point is that all models remain close to the theoretical -5/3 spectral slope. This shows that the basic physical representation of energy cascade processes stays intact despite these systematic biases.

In this sense, we will modify line 159 to: *"…models, with the ETH model generally showing the highest energy, followed by CMCC and CNRM. This is related to the large-scale offsets between the different models, that is,* differences in spectral energy levels*. While these energy levels…"*

**Technical corrections**

**1.** Acronyms should be defined throughout where they first appear, such as on lines 4 and 5. There are some acronyms that are not defined (HIRHAM, REMO,CFSR, MERRA)

A table with all the acronyms will be added to the annexes section to provide clarity and cleanliness to the main text. The table is as follows:

*Table A1. Acronyms used in the paper and their meanings*

| *Acronym* | *Meaning* |
|---|---|
| *CCLM* | *Consortium for Small-scale Modelling – Climate Limited-area Modelling.* |
| *CFSR* | *Climate Forecast System Reanalysis.* |
| *CMCC* | *Euro-Mediterranean Center on Climate Change (Fondazione CMCC).* |
| *CNRM* | *Centre National de Recherches Météorologiques (Météo-France & CNRS).* |

| | |
|---|---|
| *CNRM-ALADIN63* | *CNRM configuration of the ALADIN limited-area model, version 63 (ALADIN = Aire Limitée Adaptation dynamique Développement InterNational).* |
| *CNRM-AROME* | *CNRM configuration of AROME (Applications of Research to Operations at Mesoscale).* |
| *CORDEX-FPS* | *Coordinated Regional Climate Downscaling Experiment – Flagship Pilot Studies.* |
| *COSMO-crCLIM* | *Climate (convection-resolving) version of COSMO for climate simulations.* |
| *CPM/CPMs* | *Convection-Permitting Model(s).* |
| *ERA5* | *ECMWF Reanalysis v5.* |
| *ERA-Interim* | *ECMWF Interim Reanalysis.* |
| *ESGF* | *Earth System Grid Federation.* |
| *ETH* | *ETH Zürich — Eidgenössische Technische Hochschule Zürich (Swiss Federal Institute of Technology).* |
| *FFT* | *Fast Fourier Transform.* |
| *HIRHAM* | *Regional climate model combining HIRLAM (High-Resolution Limited Area Model) and ECHAM (ECMWF/Max-Planck model).* |
| *IMK-TRO* | *Institute of Meteorology and Climate Research – Troposphere (KIT).* |
| *JRA-55* | *Japanese 55-year Reanalysis (JMA).* |
| *KIT* | *Karlsruhe Institute of Technology.* |

| MERRA | Modern-Era Retrospective Analysis for Research and Applications. |
|---|---|
| NEWA | New European Wind Atlas. |
| NOAA GFS | National Oceanic and Atmospheric Administration – Global Forecast System (run by NCEP/NWS). |
| NWP | Numerical Weather Prediction. |
| PSD | Power spectral density. |
| RCM | Regional Climate Model. |
| REMO | REgional MOdel. |
| S(f) | Power spectral density as a function of frequency f. |
| SLHD | Semi-Lagrangian Horizontal Diffusion. |
| U50 | 50-year return-period wind speed. |
| WRF | Weather Research and Forecasting Model. |

**2.** Line 20: Is 'climatological average' the correct term here?

Thank you for spotting this. We will change *Line 20* this to *"…tail with the theoretically expected spectral slope of −5/3…"* to reflect its physical basis in a better way.

**3.** Line 91: Should be "3 km grid spacing"

Thank you for this comment. We will complement Line 91 with: *"…domain is also 3 km grid spacing, the NEWA…"*

---

## Author Comment (AC2)

**RESPONSES POINT BY POINT TO REFEREE N°2 OF WIND ENERGY SCIENCE DISCUSSIONS**
*Brief communication: Enhanced representation of the power spectra of wind speed in Convection-Permitting Models*
**Nathalia Correa-Sánchez, Xiaoli Guo Larsén, Giorgia Fosser, Eleonora Dallan, Marco Borga, and Francesco Marra**

Dear Referee No.2,

We thank you for your review work and the valuable comments, which helped to improve our paper. Our responses are reported in blue, and all the modified or new text is reported in *italics and red*. Line numbering refers to the original version of the paper that was available for the open discussion.

**General comments**

The paper deals with the issue of use of reanalysis data for wind-energy purposes. In particular, it is known that power spectrum of wind speed resulting from numerical methods suffer of energy loss at high frequencies, leading to an underestimation of extreme wind speeds needed for the structural design of wind turbine. In the manuscript, the authors evaluate the possibility of use Convention-Permitting Models to better reproduce the theoretical -5/3 slope in wind power spectrum as alternative to the most used datasets like, e.g., ERA5. The study is very interesting and contributes significantly to the important open debate on the reliability of reanalysis data for structural design purposes. The brief communication is suggested to be published after minor revision.

Thank you very much for your positive insights about our work. We will respond to each of your specific comments in the following section.

**Specific comments**

**Comment #1 - Page 3, Lines 81-83** *"Originally recorded at 10-minute intervals, these data were subsequently aggregated to hourly values by arithmetic averaging to facilitate direct comparison with the simulations of the CPM models."* Please explain whether the hourly sampled CPM data are representative of 10-min average wind speed or 1-hour average wind speed. Indeed, e.g., in ERA5 dataset the provided parameters are available hourly, and they are defined as either instantaneous value referring to a specific point-in-time (thus not averaged) or mean rate value averaged over a given time period. If the case is the first, then the "arithmetic averaging" of field measurements seems to be wrong.

We thank the referee for pointing out this relevant detail about the time window definitions of our datasets. We provide the following specifications for each dataset:

• CORDEX-FPS CPMs: Instantaneous hourly values showing the model state at each hourly timestep.

• ERA5: Instantaneous values (with an implicit 30-minute average of elapsed time).

• NEWA: Output at 30-minute temporal resolution (Hahmann et al., 2020), aggregated to hourly values for this analysis.

• Observations: Initially recorded every 4 seconds and aggregated to 10-minute means (Kohler et al., 2018), later combined into hourly mean values for comparison.

However, it is essential to note that our study examines how each dataset performs in the relevant frequency range, spanning from a few hours to ~ 1 hour, for spectral analysis. Here, we do not use this data for direct extreme wind calculations, as we stated in the introduction. At these temporal scales, different definitions of temporal windows do not greatly change our main conclusions about spectral slope behaviour.

This is because the processes that create the theoretical -5/3 slope in the mesoscale range (1-6 day^-1) work at characteristic timescales of about 4-24 hours that are much longer than the differences between instantaneous hourly values and 10-minute averages. Additionally, we confirmed this by comparing the spectral slopes between the original 10-minute observational data and the hourly-aggregated values, and found no significant differences in spectral behaviour at frequencies of ≥1 hour^-1, confirming that our comparative spectral analysis remains valid for the frequency range of interest in our study.

Therefore, while we acknowledge these methodological differences, they do not impact our primary findings regarding the superior spectral representation in CPMs compared to nudged simulations, such as NEWA, and reanalysis products, like ERA5. In this sense, we will aggregate the following paragraph in line 109:

*"All datasets are analysed at an hourly frequency for consistency and comparison, but they have different time window definitions. CPMs represent instantaneous model states. ERA5 offers instantaneous values, averaging them over 30 minutes. NEWA provides 30-minute resolution data, grouped into hourly values. Observations are collected from 10-minute averages and converted to hourly values. These differences in time windows do not influence spectral analysis in our frequency range of interest, which is from 1 to 6 day$^{-1}$, because the atmospheric processes that generate the -5/3 spectral slopes operate at much longer timescales (4-24 hours) than these methodological differences."*

**REFERENCE:** Kohler, M., Metzger, J., & Kalthoff, N. (2018). Trends in temperature and wind speed from 40 years of observations at a 200-m high meteorological tower in Southwest Germany. *International Journal of Climatology*, *38*(1), 23-34.

**Comment #2 - Page 45, lines 116-117** *"the hourly time series of CPMs, ERA5, and NEWA, were first detrended by subtracting their mean value, thus removing the constant component."* The detrending operation is usually made to remove low-frequencies oscillations that can introduce an unwanted mean component to short records. Please specify better what you mean by "detrend" in this case: is it perhaps to make time series zero-mean for the purpose of deriving power spectra?

We thank and agree with the referee's comment about the terminology. The referee is correct that 'detrended' is usually used to remove low-frequency oscillations and long-term trends over

time. In our analysis, we calculated wind speed anomalies by subtracting the average of the time series. We do this to centre the data around zero and remove the constant component needed for spectral analysis while preserving the temporal trend. We will change the term in line 116 to *'mean-centred'* to better describe the preprocessing done before spectral analysis.

**Comment #3 - Figure 1.** It is shown that CPM simulations provide enhanced spectral contribution at larger frequencies. Is there a possible physical explanation for this phenomenon?

We thank the referee for this critical question about the physical mechanisms behind the improved spectral performance of CPMs.

As we discuss throughout the original manuscript, particularly in lines 174-184 and 207-218, the better spectral representation at high frequencies in CPMs comes from fundamental differences in simulation methods that preserve natural atmospheric processes. CORDEX-FPS CPMs use continuous, free-running simulations. This setup allows convective processes to develop their own spectral characteristics without large-scale limits (Coppola et al., 2020), preserving the natural energy flow processes that create mesoscale variability, especially in the high-frequency range (1-12 day^-1) where we see spectral improvements.

On the other hand, other high-resolution models, such as NEWA, employ spectral nudging and frequent restarts (8-day runs with spectral nudging) to ensure consistency with reanalysis forcing (Hahmann et al., 2020). Vincent and Hahmann (2015) showed that spectral nudging reduces wind speed variance across all wavelengths, even at the surface where nudging is not applied. This constraint may interfere the natural energy flow processes that generate mesoscale variability, which explains the steepened spectral slopes seen in Figure 1d.

In this sense, the key difference in simulation philosophy lies between fidelity to physical processes and large-scale constraints, which explains why CPMs naturally maintain realistic high-frequency wind variability. However, the disadvantage is that these continuous simulations are computationally very costly (Coppola et al., 2020; Fosser et al., 2024).

**Comment #5** In addition to the KIT measurement site, the PSD of randomly-selected 10 locations are discussed. Since they can be relevant, please describe both roughness and orography conditions at KIT measurement site as well as at the 10 locations of Figure 2.

Thank you. To complement the information we presented in Figure A1, we will cite in line 80 the work of Kohler et al. (2018) for the description and details about the KIT mast point. In addition, we will include a table in the annexes with information on the roughness of the random points in the appendices:

*Table 1A. Characteristics of the 10 randomly selected locations for spectral analysis. Roughness length (z0) values from COSMO model and ranges based on CORINE Land Cover classifications (Demuzere et al., 2008).*

| Point | Latitude | Longitude | Elevation [m a.s.l] | zo[m] Range | Description (Examples) |
|---|---|---|---|---|---|
| Serie 1 | 40.271 | 6.1286 | 0 | 0-0.0003 | Very smooth (water, ice) |
| Serie 2 | 42.566 | 2.1006 | 1638.61 | 0.7-inf | Very rough (dense forests, urban centres) |
| Serie 3 | 42.566 | 15.5526 | 0 | 0-0.0003 | Very smooth (water, ice) |
| Serie 4 | 44.213 | 2.7086 | 813.98 | 0.3-0.7 | Rough (scattered forests, peri-urban areas) |
| Serie 5 | 46.805 | 1.0366 | 85.88 | 0.03-0.3 | Moderate (shrubs, bareland) |
| Serie 6 | 44.024 | 3.0126 | 595.05 | 0.3-0.7 | Rough (scattered forests, peri-urban areas) |
| Serie 7 | 47.426 | 4.4946 | 430.61 | 0.03-0.3 | Moderate (shrubs, bareland) |
| Serie 8 | 45.239 | 6.0906 | 2349.68 | 0.7-inf | Very rough (dense forests, urban centres) |
| Serie 9 | 40.352 | 16.2746 | 287.52 | 0.03-0.3 | Moderate (shrubs, bareland) |
| Serie 10 | 40.595 | 12.9686 | 0 | 0-0.0003 | Very smooth (water, ice) |

**REFERENCES:**
Demuzere, M., De Ridder, K., and van Lipzig, N. P. M.: Modeling the energy balance in Marseille: Sensitivity to roughness length parameterizations and thermal admittance, Journal of Geophysical Research: Atmospheres, 113, D16 120, https://doi.org/10.1029/2007JD009113,610 2008.

Kohler, M., Metzger, J., & Kalthoff, N. (2018). Trends in temperature and wind speed from 40 years of observations at a 200-m high meteorological tower in Southwest Germany. *International Journal of Climatology*, *38*(1), 23-34.

**Comment #6** The study is limited to comparison of Power Spectrum. Evaluate the possibility of compare recorded and simulated yearly maxima at KIT measurement site, e.g. by showing their empirical distribution function or, at least, the right tail of the parent distribution.

We appreciate the comment, and although the suggestion is not entirely in line with the focus and central topic of our paper on the spectral characteristics of the compared datasets, for context, we provide statistical comparisons of the datasets evaluated at the KIT point in the appendix section as supplementary information.

In line 109, we will add: *"A statistical comparison of the time series from the different datasets at the KIT point is presented in Figure A2."*

[Figure]

*Figure A2.* *Comparison of time series at the KIT point.*

---

## Author Comment (AC3)

**RESPONSES POINT BY POINT TO REFEREE N°3 OF WIND ENERGY SCIENCE DISCUSSIONS**
*Brief communication: Enhanced representation of the power spectra of wind speed in Convection-Permitting Models*
**Nathalia Correa-Sánchez, Xiaoli Guo Larsén, Giorgia Fosser, Eleonora Dallan, Marco Borga,**
**and**
**Francesco Marra**

Dear Referee No.3,

We thank you for your review work and the valuable comments, which helped to improve our paper. Our responses are reported in blue, and all the modified or new text is reported in *italics and red*. Line numbering refers to the original version of the paper that was available for the open discussion.

**General comments**

Dear authors, thanks for a short, interesting and well-written manuscript!

See my comments in the pdf attached.

I'd like you to review existing, recent works already published which adress the same topic and use CPM models. This is not to question to novelty of your work, but instead to bring you closer to the small community of CPM modellers with an interest for Wind Engineering applications (not only Wind Energy, but also Wind Hazards in general).

All the best

Rémi Gandoin, C2Wind, Denmark.

Thank you very much for your positive thoughts about our work. We will respond to each of your specific comments in the following section.

**Specific comments**

**Comment #1.** Consider adding "mean", i.e. "mean annual exceedance probability"

Thank you for this suggestion. However, we insist that *annual exceedance probability* is the correct term in extreme value statistics literature. Perhaps the reviewer meant that the return period is an average recurrence interval.

**Comment #2.** In Line 50: Please consider reviewing:

We reviewed the suggested references in the background section. Since this is a brief communication, the journal sets guidelines regarding the maximum number of references to be used, so we had to prioritise.

Chun-Hsu Su's work with the BARRA-C and BARRA-C2, and soon BARRA-3 suite of regional reanalysis https://doi.org/10.5194/gmd-14-4357-2021, their work include wind speed.

Thank you for your recommendation. However, after reviewing the suggested paper, we found that although the document is a relevant work on convection-permitting datasets and evaluates wind, it does not address the spectral properties of wind at turbine height or its direct application to 'wind energy' in detail. Therefore, it does not support the statement we made in line 50 that these areas of knowledge have not been explored.

This paper evaluates the wind speed at 10 m and the surface properties that may influence its estimation. It analyses metrics such as root mean square difference, Pearson correlation, additive bias and variance bias for wind speed. However, the paper does not discuss the spectral properties of wind, its energy spectrum or turbulence characteristics for this reanalysis. Furthermore, although it makes a general mention of the potential of reanalyses for renewable energy applications, it does not specifically explore BARRA-C's ability to reproduce wind properties in the context of wind energy applications within its own assessment. Finally, the study evaluates the BARRA-C reanalysis by focusing specifically on four mid-latitude subregions in Australia, not in central Europe.

Considering these points and the prioritisation of references that we must make within the Brief communication, we do not believe that this work offers a relevant contribution to our study. However, it does present the evaluation of a convective-scale reanalysis, which, although related, does not directly influence our scope.

Similar (and carried out in collbaration with the above authors) with the NZRA regional reanalysis (includes explicit deep convection as BARRA-C2)

https://www.data-assimilation.riken.jp/isda2024/files/pdf/p1-16.pdf
https://doi.org/10.2307/27226715

Thank you for this suggestion. However, this work, as you mention, is along the same lines as the previous one, but for New Zealand this time. Here, the study compares the performance of the NZRA for wind speed at 10 m and wind gusts, demonstrating that the NZRA better fits wind speed observations, even at higher percentile thresholds, and outperforms other reanalyses in estimating strong winds. Furthermore, the research also suggests that the knowledge generated by NZRA can contribute to various disciplines where wind energy and wind risk assessment fit perfectly. However, the study does not address the ability of CPMs to reproduce the 'spectral properties of wind'. The evaluations focus on performance metrics such as percentiles, correlation, time series, extreme event frequencies, and biases for mean wind speed and gusts.

Although this is an excellent example of the use of CPMs in meteorology and climatology to assess wind for risk and energy purposes, demonstrating significant added value in the prediction of strong winds and gusts, it does not address the spectral properties of wind and therefore its contribution focuses on the accuracy of wind magnitudes and the frequency of extremes in New Zealand, not on spectral analysis.

The following works:

https://doi.org/10.1016/j.jweia.2024.105844

Thank you for this interesting contribution. However, the source provides partial support and contextual background for the statement, but does not directly support the statement in its entirety.

The reference confirms the growing use of CPMs in meteorology and climatology and their application in the field of wind energy. It also details the limitations of these models in simulating small-scale phenomena such as gusts or grid-scale turbulence, which are aspects of wind spectral properties. However, it does not explicitly address or refer to the ability of these models to reproduce wind spectral properties in wind energy applications. Instead, it describes sub-grid scale variability as an inherent limitation of the models.

https://doi.org/10.1007/s00382-023-06803-w

After reviewing this interesting source, the conclusion is that the work of Adinolfi et al. (2023) does not fully support the claim in line 48. While it provides support for part of it, it lacks crucial information for the rest. For example, it does not evaluate or discuss the spectral properties of the wind or the ability of the VHR-REA_IT model (the same one presented in Raffa et al. (2021)) to reproduce them. Furthermore, the study focuses exclusively on the evaluation of temperature at 2 m and precipitation, not winds at any height, rather it is limited to describing how turbulent flows are parameterised in the model, but does not evaluate its performance in reproducing wind properties.

In conclusion, although this reference is very valuable because it confirms the growing interest and use of CPMs in meteorology and climatology, which we recognise due to their advantages in representing local-scale phenomena, especially temperature and precipitation, it does not provide any information or evaluation on the ability of these models to reproduce wind spectral properties, nor does it focus on the specific context of wind energy applications.

https://journals.ametsoc.org/view/journals/apme/60/10/JAMC-D-21-0029.1.xml (NORA3)

Thank you for this other interesting work. Although the purpose of the study is to demonstrate that NORA3 (a reanalysis covering mainly Norway and other regions of northern Europe) significantly improves the wind field compared to previous reanalyses such as ERA5 and NORA10, especially in mountainous areas and along coastlines with enhanced grid resolution which si very relevant for wind energy; as in the previous recommendations, the article does not explicitly discuss or evaluate the spectral properties of wind, which are the fundamental objective of our Brief Communication and what we refer to. Therefore, we do not consider that this work, although very valuable for other related topics, supports the idea we want to convey in line 49, to which the comment refers.

I believe all the above references uses CPM for wind-related analyses.

Thank you very much for your suggestions. It is indeed very interesting literature that we enjoy reviewing, but in the case of the statement in question (line 49), while we do not consider it to contribute significantly as a back ground in the spectral characteristics of CPM simulations of wind speeds at 100 m. Moreover, we needed to consider the limit on the references allowed in WES for a Brief Communication. However, based on your comment, we have included two of the

most recent references you suggested that support the use of wind fields from CPM datasets to feed the first part of the sentence, although they do not cover spectral characteristics. That is why we will change line 49 as follows:

*"…Despite the increasing use of CPMs in meteorology and climatology* (Pirooz et al., 2023; Raffaele et al., 2024)*, their ability to reproduce wind spectral properties in the context of wind energy applications have not yet been explored in detail…"*

**Comment #3.** In th observational data subsection: A reference to a document describing the measurements (type of sensors, mounting etc) needs to be provided.

Thank you. We will add the reference *"Kohler et al. (2018)"* of the scientific paper describing the mast observation in Line 80.

**REFERENCE:** Kohler, M., Metzger, J., & Kalthoff, N. (2018). Trends in temperature and wind speed from 40 years of observations at a 200-m high meteorological tower in Southwest Germany. *International Journal of Climatology*, *38*(1), 23-34.

**Comment #4.** I think you can find others, with measurements closer to the surface, such as

W1M3A    http://www.w1m3a.cnr.it/OI1/modules/site_pages/about.php

You could discuss why not using measurements closer to the surface, and easier to find; possibly w references to the two Italian papers I mentioned in my earlier comment, one of them uses measurements from 21 stations.

We appreciate the reviewer's suggestion regarding W1M3A and other surface measurement networks. We chose the KIT mast dataset after a thorough an exhaustive search for open-access wind measurements that met strict criteria: (1) spatial coverage within our study area, (2) at least 10 years of data for solid spectral analysis, (3) hourly resolution to match our model outputs, and (4) measurements at 100m height to avoid vertical extrapolation errors.

After an examination of the W1M3A observatory, we noticed that meteorological measurements are taken at about 7-15 meters above sea level on the upper mast of the ODAS Italia 1 spar buoy. This height is much lower than our study's target of 100m. Moreover, W1M3A is located offshore in the Ligurian Sea, around 80 km from the coast.

Extrapolating measurements from around 10 m in a marine setting to 100m would lead to several systematic uncertainties. These include assumptions about vertical extrapolation, the differences in atmospheric stability over the ocean compared to land, and the boundary layer characteristics. Additionally, the transition from marine to terrestrial conditions would add more bias when comparing with our mainly terrestrial CPM grid points.

Our method focuses on eliminating factors that could be misinterpreted as differences in CPM spectral performance. Using measurements taken at the model native output height of 100 m allows for the cleanest evaluation of spectral accuracy without the errors associated with extrapolation or differences in environmental conditions.

While the W1M3A and Italian networks are important meteorological resources, they do not meet our specific needs for an unbiased CPM spectral evaluation at wind turbine hub heights.

**Comment #5. (in Line 98)** Other studies such as https://iopscience.iop.org/article/10.1088/1742-6596/2151/1/012009 looked into this, consider referring to them.

If you end up quoting the NORA3 paper, see this study where they look at both wind speed and precipiation spectra https://doi.org/10.1016/j.rineng.2024.102010, it would be good to refer to it.

Thank you for your recommendation. Like Bastine et al. (2018), which we already included in the text, Meyer et al. (2022) also cover and apply spectral corrections to NEWA data (and ERA5 data). Both articles use and evaluate the spectral correction applied to NEWA data with the aim of obtaining more accurate estimates of extreme winds. The relevance for us would be that, as with Bastine et al. (2018), this work also recognises that mesoscale simulations such as NEWA tend to smooth out high-frequency wind fluctuations. To this end, we have changed the current sentence in line 98:

*"…However, this evaluation did not specifically address the spectral characteristics or the representation of high-frequency variability that we examine in this study…"*

For the following sentence:

*"…However,* Bastine et al. (2018) and Meyer et al. (2022) *applied spectral corrections to address the smoothing effect and correct the underestimation of extreme winds, since they detected that NEWA tends to smooth out high-frequency wind fluctuations…"*

**Comment #6.** About ERA5: I suggest to mention that it used as forcing for NEWA.

Thank you very much for bringing this important detail to our attention. Yes, we will indeed add the following to line 109 to close that paragraph:

*"…and wind energy applications. Furthermore, ERA5 was the main source of initial and boundary conditions for the NEWA simulations."*

**Technical corrections**

Thank you for the detailed review and for correcting the typos and spelling mistakes. We will correct all the ones you have pointed out, and we will also review the rest of the manuscript once again to ensure that everything is spelt correctly.

---

## Author Response (AR3)

**Responses to final comments from the Associated Editor**

Dear Editor-in-Chief Dr Julia Gottschall and Associate Editor Dr Etienne Cheynet,

Thank you very much for your positive feedback on our research.

Below we respond to the points raised by the Associate Editor with technical corrections in blue, which are very useful for the final refinement of our work.

**Technical corrections:**

- Regarding "instantaneous values" in lines 120–127: Wind data from reanalysis products (e.g., ERA5) represent mean values from a Reynolds-averaged model, not true instantaneous winds. Although some variables are labelled "instantaneous," turbulence has already been parameterized, so I am afraid that describing them as instantaneous can be misleading.

Thank you for pointing this out. We have corrected section 2. Data in lines 120 and 121 of the file with tracked changes to incorporate this important methodological clarification.

- For spectral analysis, it is advisable to remove both the mean and the linear trend from the time series. Detrending helps prevent spectral leakage and distortion in the low-frequency range, which can occur if only the mean is removed.

Thank you for pointing this out. We have taken this comment into account and applied the *signal.detrend* function in Python to remove the linear trend from the data already centred on the mean. We have also specified this in line 134. We have noted that, in our case, this trend is not significant, as the spectra are identical to how they were before. To confirm, we plotted the trend lines of the time series, which are practically horizontal with a slope of zero.

- A LaTeX reference appears to be broken on line 27 ("(?)").

Thank you, the broken reference symbol (?) that appeared in the previous change tracking file is no longer in the clean version of the manuscript.

Many thanks again to the Associate Editor for his detailed review of our work; his comments have improved our work.